# Revealing an Overlooked Challenge in Class-Incremental Graph Learning

**Daiqing Qi**                                                          *daiqing.qi@virginia.edu*
*University of Virginia*
*Charlottesville, VA 22903*

**Handong Zhao**                                                        *hazhao@adobe.com*
*Adobe Research*
*San Jose, CA 95110*

**Xiaowei Jia**                                                         *xiaowei@pitt.edu*
*University of Pittsburgh*
*Pittsburgh, PA 15260*

**Sheng Li**                                                           *shengli@virginia.edu*
*University of Virginia*
*Charlottesville, VA 22903*

**Reviewed on OpenReview:** *https://openreview.net/forum?id=ScAc73Y1oJ*

## Abstract

Graph Neural Networks (GNNs), which effectively learn from static graph-structured data, become ineffective when directly applied to streaming data in a continual learning (CL) scenario. In CL, historical data are not available during the current stage due to a number of reasons, such as limited storage, GDPR[1] data retention policy, to name a few. A few recent works study this problem, however, they overlook the uniqueness of continual graph learning (CGL), compared to well-studied continual image classification: the unavailability of previous training data further poses challenges to inference in CGL, in additional to the well-known catastrophic forgetting problem. While existing works make a strong assumption that full access of historical data is unavailable during training but provided during inference, which potentially ***contradicts*** the continual learning paradigm (Van de Ven & Tolias, 2019), we study continual graph learning without this strong and contradictory assumption. In this case, without being re-inserted into previous training graphs for inference, streaming test nodes are often more sparsely connected, which makes the inference more difficult due to insufficient neighborhood information. In this work, we propose ReplayGNN (ReGNN) to jointly solve the above two challenges without memory buffers: catastrophic forgetting and poor neighbor information during inference. Extensive experiments demonstrate the effectiveness of our model over baseline models and its effectiveness in different cases with *different* levels of neighbor information available.

## 1 Introduction

Graph Neural Networks (GNNs) (Kipf & Welling, 2016a; Hamilton et al., 2017) have been recognized as a valid tool for graph learning, showing promising performance on a variety of tasks. In practical scenarios,

---

[1]https://en.wikipedia.org/wiki/General_Data_Protection_Regulation

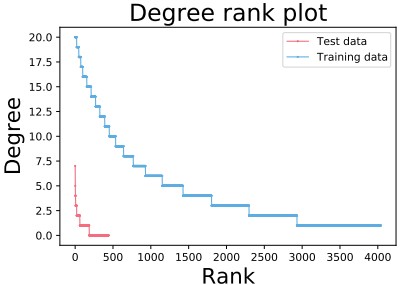 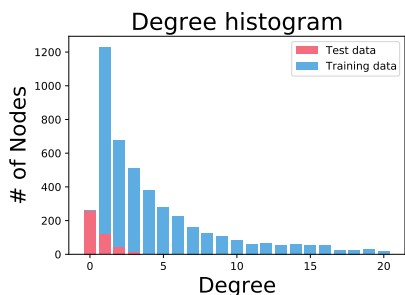

Figure 1: Node degree statistics of training and test data from a graph that records user interactions in Stack Exchange (an online platform). We extract data accumulated for a month as the training graph and data from the following few days as the test data. Obvious difference between training and test data *density* can be observed under continual graph learning setting (previous training nodes are unavailable).

graphs can often evolve over time, and meanwhile, previous data (e.g., some nodes and edges) are inaccessible sometimes. For instance, in user interaction graphs of online platforms (Fig. 1), history records can be unavailable due to customer data storage consent, for instance, the 30-days right-of-erasure in General Data Protection Regulation (GDPR), or the users' preferences to hide their history records. Traditional graph learning models (Hamilton et al., 2017; Kipf & Welling, 2016a), which are not designed for preserving good performance on all learnt tasks over time, often have limited performance in this setting because the unavailability of previous data leads to catastrophic forgetting (McCloskey & Cohen, 1989).

Continual learning (Thrun, 1995) aims to develop an intelligent system that can continuously learn from new tasks without forgetting learnt knowledge in the absence of previous data. Common continual learning scenarios can be roughly divided into two categories (Van de Ven & Tolias, 2019): task-incremental learning (TI) and class-incremental learning (CI) (Rebuffi et al., 2017; Qi et al., 2023). Recently, a few works (Liu et al., 2021; Zhou & Cao, 2021; Wang et al., 2020; 2022; Galke et al., 2021) begin to study Continual Graph Leaning (CGL), which is helpful when accumulating all history data over time is not feasible due to storage pressure or customer data storage consent, for instance, the 30-days right-of-erasure in GDPR. [2]. Furthermore, even in scenarios where all prior data remains accessible, continual graph learning still holds significance. This is because it is advantageous for the model to incrementally learn from new data, rather than undergoing a complete retraining process from scratch.

However, in the context of graph learning, the unavailability of previous data further poses a unique challenge in addition to the catastrophic forgetting problem during the inference. Specifically, existing graph-based models often consider nodes in both training and testing data when performing graph operations in the evaluation phase, e.g., aggregating neighbors for test nodes. However, this is not practical in the continual learning setting because a subset of the nodes and edges are from previous tasks and they may not be available for the following tasks.

When existing works evaluate their models under the continual graph learning setting with citation graphs, social networks, or co-purchase graphs, during the inference, they re-insert test nodes into the previous training graph and then aggregate the neighborhood information of test nodes to make a prediction (Fig. 2). Such evaluation method is common in a traditional graph learning setting. Nevertheless, in continual graph learning, it is not always practical. This is because in continual learning, training data (previous nodes) from previous tasks are no longer available at the current stage, including both training and *inference* stage. In a typical continual graph learning scenario, on-the-fly inference is often required. For instance, in node classification, the model needs to infer the labels of streaming test nodes once it receives them, instead of accumulating a large number of test nodes to do an inference. Without re-inserting back to the graph, the streaming test nodes of a relatively smaller size are often very sparsely connected. It means while the model is trained on a dense graph, it is required to infer on a much sparser graph. Fig. 1 showcases the above point

---

[2]https://gdpr-info.eu/

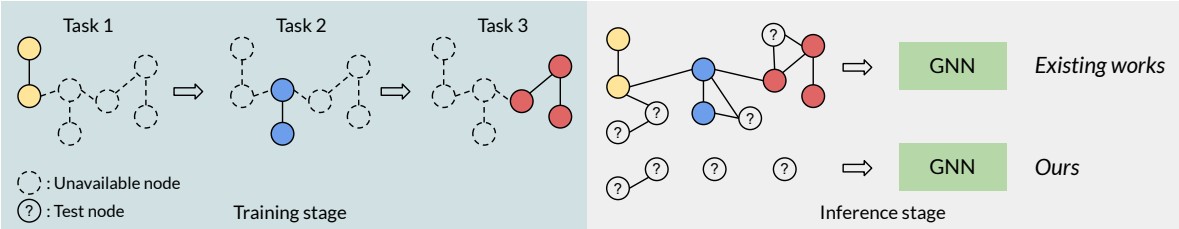

Figure 2: Different cases during the *inference* stage. Compared with existing works (Liu et al., 2021; Zhou & Cao, 2021; Wang et al., 2022), ours is more practical and strictly follows the continual learning paradigm: previous training nodes and edges are no longer available during current stage, which includes both the training stage and the **inference** stage.

in a real-world scenario. We study with the user interaction graph from the Stack Exchange platform [3] and observe notable difference between training and test data density under the continual graph learning setting. The above facts lead to a key challenge in addition to catastrophic forgetting in continual graph learning: *the neighborhood information available during the inference is very limited.* Different from existing works, we consider this practical yet ignored case in our paper and solve this challenge.

Besides, different from existing works that either focus on task-incremental learning and data-incremental learning, or focus on class-incremental learning but use a buffer to store previous data for replay to prevent forgetting, in this paper, we focus on CI graph learning problem *without* memory buffers. In light of the unavailability of history data, we are motivated by the success of generative replay based models in image classification tasks and further propose a generative replay based GNN model, ReplayGNN (ReGNN), to solve the above challenges. Without requiring storing previous data, it outperforms baseline models including competitive baselines with memory buffers on several benchmark graph datasets.

The main contributions of this paper are as follows: (1) We find an interesting and important yet overlooked fact by existing works on continual graph learning: the unavailability of previous training nodes and edges further lead to the poor neighborhood information problem during the *inference* stage, in addition to the catastrophic forgetting problem. (2) Taking this ignored fact into account, we study the class-incremental graph learning problem *without* the use of memory buffers. (3) We further propose ReplayGNN to address the problems and the experiments demonstrate its effectiveness in different cases given *different* levels of neighbor information.

## 2 Related Work

### 2.1 Graph Neural Networks

Graph neural networks are popular models for graph representation learning. The success of GNNs has boosted research on many real world tasks based on graph-structured data, ranging from node classification(Jiang et al., 2019; Rezayi et al., 2021; Jiang et al., 2023; Shi et al., 2023), knowledge graph based question answering (Hua et al., 2020), to recommender systems (Sheu et al., 2022). GNNs can be roughly categorized into two groups: spatial methods (Hamilton et al., 2017) and spectral methods (Kipf & Welling, 2016a). GraphSAGE (Hamilton et al., 2017) is a typical spatial method, which directly aggregates node representations from a node's neighborhoods to obtain its representation, while graph convolutional network (GCN) (Kipf & Welling, 2016a), a typical spectral method, learns graph representation in the spectral domain, which alleviates over-fitting on local neighbors via the Chebyshev expansion. To further effectively learn from the local structure, graph attention networks (GAT) (Velickovic et al., 2017) adopt an attention mechanism based on GCN. Both classic spatial and spectral methods need the entire graph during model training. To learn more effectively when scaling to large evolving graphs. Some sampling strategies have

---

[3]https://stackexchange.com/

been developed (Hamilton et al., 2017; Chen et al., 2018) so that only part of the graph is required for each iteration during training.

Although the GNNs have demonstrated their superiority in many graph-related tasks, most of them focus on graph learning on static graph-structured data. However, in real world applications, graphs are often evolving, either in form of data or classes, and history data might be no longer available due to privacy or storage concerns. Traditional GNNs often fail in such cases due to catastrophic forgetting. The performance on previous tasks would drop significantly, if the standard GNNs are applied in this setting without any modification.

## 2.2  Continual Learning

Continual learning studies the problem of learning from streaming data, with the aim of continuously acquiring new knowledge while maintaining its learnt knowledge. Current most studied continual learning settings can be roughly categorized into two categories (Van de Ven & Tolias, 2019) :Task-Incremental Learning (TI), and Class-Incremental Learning (CI). Note that in both TI and CI, number of classes can be increased as the number of task increases. The key difference is that, in TI, the task to which a test sample belongs is provided during the inference (i.e. the task-ID is available during inference). The change of data distribution can stem from different domains (Qi et al., 2024; Zhu et al., 2024; 2023) (i.e., domain-incremental), e.g., from the classification of handwritten digital numbers in MNIST (Deng, 2012) to the classification of street view house numbers (Netzer et al., 2011). The change of data distribution can also be caused by different tasks (i.e., task-incremental), e.g., from classifying handwritten digital numbers 0-4 (*Task 1*) to 5-9 (*Task 2*) in MNIST. A more challenging setting is class-incremental learning, where the data distribution shift is the result of new incoming classes and the model is required to classify both the previously learnt classes and new classes in all tasks at one time. While in task-incremental learning, the model needs to distinguish classes within each task but is not required to distinguish classes in different tasks because task-IDs are available.

Class-incremental learning (CI) (Rebuffi et al., 2017) is a harder continual learning problem due to the unavailability of task-IDs. Representative continual learning models, such as (Kirkpatrick et al., 2017) and (Li & Hoiem, 2017), achieve promising performance on TI benchmarks, but suffer from notably forgetting on simple CI benchmarks. Existing approaches to solve the CI problem can be divided into three categories (Ebrahimi et al., 2020), including the replay-based methods (Rolnick et al., 2019; Chaudhry et al., 2019), structure-based methods (Yoon et al., 2017), and regularization-based methods (Kirkpatrick et al., 2017; Aljundi et al., 2018). Progress has been made in continual learning in recent years, however, only a few of them study continual learning with GNNs.

## 2.3  Continual Graph Learning

Different continual graph learning settings are explored in recent studies, including Data Incremental Learning (DI) (Wang et al., 2020; Xu et al., 2020; Cai et al., 2022; Han et al., 2020; Wang et al., 2022), Task-Incremental Learning (TI) (Liu et al., 2021; Zhou & Cao, 2021; Zhang et al., 2021) and Class-Incremental Learning (CI)(Wang et al., 2022). (Cai et al., 2022) studies continual learning in a multi-modal graph with neural architecture search, and (Xu et al., 2020) prevents the recommendation system from forgetting the long term user preference by knowledge distillation. (Daruna et al., 2021) and (Kou et al., 2020) focus on the continual learning of knowledge graph embeddings. Their methods can be further considered in two scenarios by if they use a buffer to store raw data (Zhou & Cao, 2021) or prototypes (Zhang et al., 2021). The most related work to ours includes (Zhou & Cao, 2021), (Liu et al., 2021) (Wang et al., 2022) and (Zhang et al., 2021). DI is not a typical continual learning setting, but it is studied in the context of graph learning. In DI, all samples are streamed randomly, while in CI and TI, all samples from a group of classes are streamed before switching to the next group. TWP (Liu et al., 2021) is a regularization-based method without storing any data or prototypes while (Zhou & Cao, 2021), (Zhang et al., 2021) and (Wang et al., 2022) need a buffer to store either raw data or prototypes to prevent forgetting.

However, existing works overlook an interesting and important fact that is unique in the context of continual graph learning compared to regular continual learning in image classification: the unavailability of previous

training nodes and edges lead to limited neighborhood information during *inference*, in addition to the catastrophic forgetting. Our work differs from existing works in that 1. we consider this ignored fact and 2. we focus on CI setting without the use of a memory buffer to store raw data or prototypes.

## 3 Methodology

### 3.1 Problem Formulation

**Class-Incremental Graph Learning** is defined as follows. Denote a graph as $\mathcal{G} = \{\mathcal{V}, \mathcal{E}\}$. A model learns from a sequence of data $D = \{\mathcal{D}^1, \mathcal{D}^2, ..., \mathcal{D}^m\}$, where $\mathcal{D}^i = \{\mathcal{V}^i, \mathcal{E}^i\}$. Each $\mathcal{D}^i$ is the data distribution of the corresponding task $\mathcal{T}^i$, with $\mathcal{Y}^i$ being its label space. When performing the task $\mathcal{T}^t$, we assume access to only $\mathcal{D}^t$, i.e., all data $\{\mathcal{D}^i | i < t\}$ is unavailable. The goal is to effectively learn from $\mathcal{D}^t$, while maintaining the model performance on learnt tasks. At the end of $\mathcal{T}^m$ (i.e., all $m$ tasks are learnt), the model is required to map test samples from *all* seen data distributions to $\mathcal{Y}^1 \cup \mathcal{Y}^2 ... \cup \mathcal{Y}^m$ without task-IDs. *Buffers to store raw data or prototypes are not allowed.* More discussions about out setting is available in Appendix A.

### 3.2 An Ignored Fact During Inference Stage

In typical image classification scenarios, training data are not required for inference because test samples are assumed to be independent of training samples. In contrast, graph learning often leverages the dependencies between test nodes and training nodes for better inference when learning with large networks. For instance, in many cases, test nodes are re-inserted into the training graph to exploit the dependency by neighbor aggregation for better inference. However, most of existing graph learning models cannot be applied to the continual graph learning scenario as the data from previous tasks become unavailable for inference. Besides, directly applying standard graph models often lead to the catastrophic forgetting problem over previous tasks.

However, existing works (Wang et al., 2022; Liu et al., 2021; Zhou & Cao, 2021) on TI and CI graph learning overlook the challenge of unavailable previous data during inference. Fig. 2 illustrates different inference cases in inductive node classification task. We do not consider transductive learning because it requires all test samples to appear in the graph for training from the beginning, which is not practical in CI graph learning, where streaming test samples often appear over time. The scenario adopted by existing works is not practical because in continual learning, data (i.e. nodes and edges) from previous tasks are no longer available once learned. However, in existing works, test nodes with labels from all learnt classes are re-inserted into the graph and connected to previous nodes for inference.

Considering the unavailability of previous training data, we introduce ours as the practical inference scenario in continual graph learning, where previous training nodes are unavailable and only connections among test nodes are kept. In this case, as discussed in section 1, the connections among test nodes are very sparse. It leads to a unique challenge in graph continual learning: the neighborhood information available for inference is very poor.

### 3.3 ReGNN

To solve the major challenges in continual graph learning: (1) catastrophic forgetting and (2) limited neighborhood information during the inference stage, we propose our model ReplayGNN (ReGNN). Fig. 3 illustrates how replay data is generated and used and Fig. 4 illustrates the detailed framework of ReGNN. The proposed GraphCVAE module is maintained to generate old data for replay to prevent forgetting, while the NodeAE module effectively learns from node attributes with little or no neighborhood information to adapt to our introduced inference case in Fig. 2.

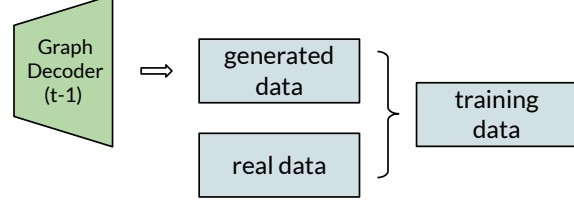

Figure 3: Composition of training data for current task $t$. It consists of replay data generated by the copy of the graph encoder from the last task $t-1$ and the training data from the current task $t$.

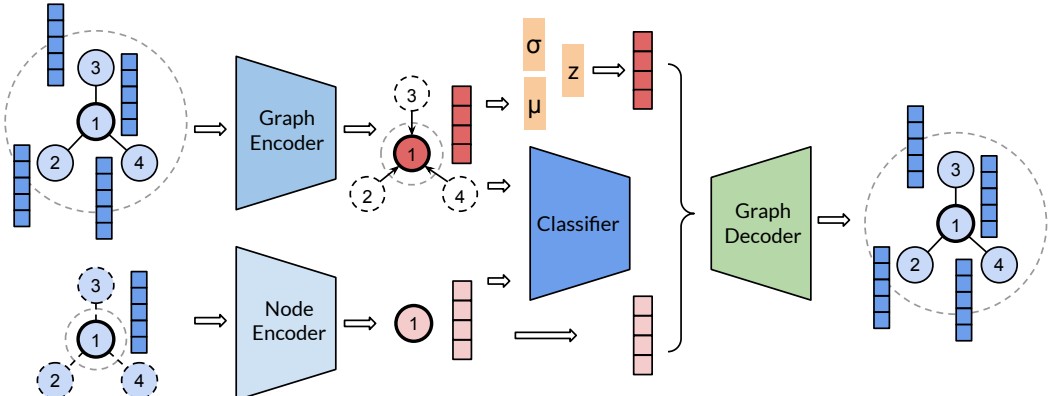

Figure 4: Overview of our ReplayGNN (ReGNN) model. ReGNN consists of three modules with shared components: a GraphCVAE, a classifier and a NodeAE. The input in the figure is a node ($node_1$) in a mini-batch with its sampled neighbors. The graph encoder consists of the convolution layer of GNN, which outputs low-dimensional embeddings (red) of $node_1$. The **GraphCVAE** module consists of the graph encoder and graph decoder. The **classifier** is a single classification layer. The **NodeAE**, which consists of the node encoder and the graph decoder (shared with GraphCVAE), takes a single node embedding as input and (1) tries to reconstruct the structure information and (2) yield node embedding (pink) that is distinguishable for classifier.

Conceptually, GraphCVAE and NodeAE are designed for two different purposes corresponding to two different challenges in CGL, but they are inherently associated through a shared graph decoder, which further boosts the model performance as ablation study illustrates.

We use a single-layer GNN in our model for simplicity. More advanced graph generation techniques can be integrated into our graph encoder and decoder if multiple layers are required. Experiments show our lightweight model is already effective. More details are discussed in the future work part in Section 5.

### 3.3.1 GraphCVAE

Based on VAE (Kingma & Welling, 2013) (details are available in Appendix A), we propose a Graph Conditional Variational Autoencoder (GraphCVAE) module, which consists of the graph decoder and graph encoder. During task $t$, GraphCVAE is trained with real data $\mathcal{D}^t$ and generated replay data $\mathcal{D}_g$, which is generated by the graph decoder $\mathrm{D}_{graph}^{t-1}$. The generation is conditioned on class labels, which contains all classes learnt before the current task. Different from Variational Graph Auto-Encoders (Kipf & Welling, 2016b), which reconstructs the whole adjacency matrix of a graph without node attributes (for link prediction during inference), our GraphCVAE tries to predict the node attributes for the neighbors of a target node given its attributes.

**Graph Encoder.** Given input nodes $\mathcal{D}^t$ with sampled neighbors, the graph encoder first obtains the node embeddings with the forward propagation steps. Denote the input node as $v$ and $\boldsymbol{h}_u^{in}$ as the raw features of a node $u$. The output embedding $\boldsymbol{h}_v^{out}$ is calculated by:

$$\boldsymbol{h}_{\mathcal{N}(v)}^{out} \leftarrow \mathrm{AGGREGATE}\left(\left\{\boldsymbol{h}_u^{in}, \forall u \in \mathcal{N}(v)\right\}\right), \tag{1}$$

$$\boldsymbol{h}_v^{out} \leftarrow \varphi\left(\boldsymbol{W} \cdot \mathrm{CONCAT}\left(\boldsymbol{h}_v^{in}, \boldsymbol{h}_{\mathcal{N}(v)}^{out}\right)\right), \tag{2}$$

where $\boldsymbol{W}$ is the learnable parameter and $\varphi$ is the activation function. Similar to the standard VAE, the mean $\mathbf{z}_\mu$ and standard deviation $\mathbf{z}_\sigma$ of the distribution are calculated from the embedding $\boldsymbol{h}_v^{out}$ by:

$$\boldsymbol{z}_\mu = \mathrm{ReLU}(\boldsymbol{W}_\mu \cdot \boldsymbol{h}_v^{out}), \; \boldsymbol{z}_\sigma = \mathrm{ReLU}(\boldsymbol{W}_\sigma \cdot \boldsymbol{h}_v^{out}), \tag{3}$$

where Rectified Linear Unit (ReLU) (Nair & Hinton, 2010) is the activation function. $\boldsymbol{W}_\mu$ and $\boldsymbol{W}_\sigma$ are learnable parameters. With mean $\boldsymbol{z}_\mu$ and standard deviation $\boldsymbol{z}_\sigma$, latent variable $\mathbf{z}$ is sampled from the Gaussian distribution $N(\boldsymbol{z}_\mu, \boldsymbol{z}_\sigma)$.

**Graph Decoder.** The decoder tries to reconstruct the input data from the latent variable $\mathbf{z}$. It first generates embedding $\boldsymbol{f}_v$ from latent variable $\mathbf{z}$ with learnable parameter $\boldsymbol{B}$, then the input embedding is reconstructed:

$$\boldsymbol{f}_v = \varphi(\boldsymbol{B} \cdot \mathbf{z}). \tag{4}$$

$$[\boldsymbol{f}_v^{out}; \boldsymbol{f}_{\mathcal{N}(v)}^{out}] \leftarrow \varphi(\boldsymbol{M} \cdot \boldsymbol{f}_v), \tag{5}$$

where $\boldsymbol{M}$ is the learnable parameter and $[\mathbf{a}; \mathbf{b}]$ is the concatenation of embedding $\mathbf{a}$ and $\mathbf{b}$. The decoded embeddings $[\boldsymbol{f}_v^{out}; \boldsymbol{f}_{\mathcal{N}(v)}^{out}]$ try to match the input embedding $[\boldsymbol{h}_v^{in}, \boldsymbol{h}_{\mathcal{N}(v)}^{out}]$, which is the return value of the CONCAT function in 1. This embedding contains the neighbor information of $v$ as well as the feature of the node $v$ itself. We do ***not*** further decode it to obtain $\{\boldsymbol{h}_u^{in}, \forall u \in \mathcal{N}(v)\}$ because the current decoded value already can be the input of graph encoder and be helpful for replay. In other words, we replay the input embedding $[\boldsymbol{h}_v^{in}, \boldsymbol{h}_{\mathcal{N}(v)}^{out}]$ instead of $\{\boldsymbol{h}_u^{in}, \forall u \in \mathcal{N}(v)\}$. Then the loss functions in GraphCVAE are:

$$\mathcal{L}^{Recon} = \text{DIST}([\boldsymbol{f}_v^{out}; \boldsymbol{f}_{\mathcal{N}(v)}^{out}], [\boldsymbol{h}_v^{in}; \boldsymbol{h}_{\mathcal{N}(v)}^{out})]), \tag{6}$$

$$\mathcal{L}^{KLD} = D_{KL}[\mathcal{N}(\boldsymbol{z}_\mu, \boldsymbol{z}_\sigma) || N(\boldsymbol{c}, 1)], \tag{7}$$

where DIST measures the distance of the two embeddings. Specifically, binary cross entropy is used as the measure of distance here. and $D_{KL}$ is the Kullback–Leibler divergence. $\boldsymbol{c}$ is the class label of input data. The overall loss function is:

$$\mathcal{L}^{cvae} = \mathcal{L}^{Recon} + \mathcal{L}^{KLD}. \tag{8}$$

### 3.4 NodeAE

Node Autoencoder (NodeAE) consists of the an encoder and a graph decoder. During the training, given a node $v$, as graphSAGE aggregates the neighbors of $v$ to obtain its representation, the classifier learns to classify $\boldsymbol{h}_v^{out}$ in Eq. 1, i.e., $\varphi(\boldsymbol{W} \cdot [\boldsymbol{h}_v^{in}; \boldsymbol{h}_{\mathcal{N}(v)}^{out}])$. However, during the inference stage, the neighbors of $v$ can be very sparse (2). In this case, the classifier fails to fully exploit its learnt knowledge with rich neighbor information on training graph for inference. Thus during the inference, given a node $v$ with only few neighbors, the extracted feature of this node, which is fed to the classification layer, is also expected to contain the neighbor information of the node. In this way, the node can be better classified by the classifier. We use an autoencoder to achieve it.

Given a node $v$, while the graph encoder takes node $v$ and its sampled neighbors as input, node encoder ***only*** takes the node attribute of node $v$ as input. But the decoder is required to reconstruct the embeddings of the aggregation of its neighbors as well as its own embedding. Another option is to generate random masks to mask certain portion of neighbor nodes of the input node before the neighbor aggregation process, while the decoder also wants to do the reconstruction. Here we simply mask all neighbors and only use the input node attribute, which is still effective. The forward propagation of the encoder is:

$$\boldsymbol{x}_v^{out} \leftarrow \varphi(\boldsymbol{H} \cdot [\boldsymbol{h}_v^{in}; ]), \tag{9}$$

where $\boldsymbol{H}$ is the learnable parameter, whose size is the same as $\boldsymbol{W}$. The loss function is:

$$\mathcal{L}^{ae} = \text{DIST}(\varphi(\boldsymbol{M} \cdot \boldsymbol{x}_v^{out}), [\boldsymbol{h}_v^{in}; \boldsymbol{h}_{\mathcal{N}(v)}^{out}]). \tag{10}$$

When trained properly, given an isolated node, the node encoder can map the node attribute into a compressed embedding $\boldsymbol{x}_v^{out}$ that contains its neighbor information and is then used for prediction.

Table 1: Accuracy on different datasets under class-incremental learning scenario.

| Model | Scenario | | Dataset | | |
|---|---|---|---|---|---|
| | No Memory Buffer | Memory size | Citeseer | Cora | Amazon |
| ER-GNN (Zhou & Cao, 2021) | ✗ | M = 1% | $34.13 \pm 0.14$ | $41.78 \pm 0.62$ | $50.53 \pm 2.94$ |
| | | M = 3% | $41.46 \pm 2.32$ | $51.96 \pm 1.65$ | $60.53 \pm 2.85$ |
| | | M = 5% | $45.48 \pm 2.35$ | $57.96 \pm 2.58$ | $67.82 \pm 2.23$ |
| ContinualGNN (Wang et al., 2020) | ✗ | M = 1% | $35.94 \pm 0.28$ | $38.48 \pm 0.17$ | $49.53 \pm 1.74$ |
| | | M = 3% | $46.08 \pm 0.45$ | $46.49 \pm 0.51$ | $56.78 \pm 2.04$ |
| | | M = 5% | $50.60 \pm 0.49$ | $60.41 \pm 1.91$ | $60.79 \pm 2.23$ |
| GraphSAGE (Hamilton et al., 2017) | ✓ | - | $31.02 \pm 0.12$ | $25.00 \pm 0.14$ | $37.01 \pm 0.43$ |
| LwF (Li & Hoiem, 2017) | ✓ | - | $36.74 \pm 1.15$ | $42.27 \pm 1.54$ | $50.01 \pm 1.83$ |
| ReGNN (Ours) | ✓ | - | $\mathbf{55.02} \pm 2.01$ | $\mathbf{64.33} \pm 1.82$ | $\mathbf{69.32} \pm 1.48$ |

## 3.5 Classifier

The classifier is a single classification layer and takes the output of the graph encoder and the node encoder as input. The loss function for classification is:

$$\mathcal{L}^{cls} = L_{ce}(g(\mathbf{x}_v^{out}, \boldsymbol{h}_v^{out}), \boldsymbol{y}), \tag{11}$$

where $L_{ce}$ is the cross entropy loss function, and $g(\cdot)$ is a function that combines two embeddings, and we simply add them together. Note that the current training data are a mixture of real data and replay data as 3 illustrates. $\boldsymbol{y}$ are their corresponding labels.

By summarizing the three modules together, the overall loss function for our ReGNN model is:

$$\mathcal{L}^{ReGNN} = \mathcal{L}^{cls} + \mathcal{L}^{cvae} + \mathcal{L}^{ae}. \tag{12}$$

# 4 Experiments

## 4.1 Datasets

Following related works (Wang et al., 2022; Liu et al., 2021; Zhou & Cao, 2021), we conduct experiments on three benchmark datasets under the continual learning settings: **Cora** (Sen et al., 2008), **CiteSeer** (Sen et al., 2008), and **Amazon** (McAuley et al., 2015), which is a segment of Amazon co-purchasing graph. We split each dataset into several tasks, following (Liu et al., 2021; Zhou & Cao, 2021; Wang et al., 2022). **CoraFull (McCallum et al., 2000)** and **Arxiv (Hu et al., 2020)** are both citation networks. **Reddit (Hamilton et al., 2017)** is a post-to-post graph. The **Products (Hu et al., 2020)** dataset is the co-purchase network. Details about dataset information and how we build tasks for continual learning scenario are available in Appendix A.

In our class-incremental setting, we divide Cora into three tasks. The first and second tasks consist of two classes, and the last task has three classes. Citeseer are split into three tasks with two classes in each task. The first task of Amazon has two classes. The second and third tasks on Amazon dataset have three classes. The model is required to classify all learnt classes *without* using task-IDs.

## 4.2 Baselines

We adopt the following baselines in our experiments. **GraphSAGE** (Hamilton et al., 2017) is a representative GNN model, which is also used as the backbone for some other baselines for fair comparison. **LwF** (Li & Hoiem, 2017) is a representative data-free continual learning model, which uses the history state of itself as the teacher for knowledge distillation to avoid forgetting. **ER-GNN** (Zhou & Cao, 2021) uses a buffer to store data for experience replay, in order to prevent from forgetting. **ContinualGNN** (Wang et al., 2020) alleviates forgetting by storing data for replay and adopts a model regularization similar to EWC (Kirkpatrick et al., 2017). **Finetuning** is the lower bound baseline by updating the model only with newly incoming

Table 2: Accuracy on Citeseer dataset at different levels of neighborhood information. Best results in bold.

| Model | Default | 3-hop | 5-hop | Full |
|---|---|---|---|---|
| ER-GNN (M=1%) | $34.13 \pm 0.14$ | $36.44 \pm 0.42$ | $36.64 \pm 0.25$ | $38.65 \pm 0.93$ |
| ER-GNN (M=3%) | $41.46 \pm 2.32$ | $49.29 \pm 2.76$ | $50.20 \pm 2.23$ | $58.82 \pm 3.32$ |
| ER-GNN (M=5%) | $45.48 \pm 2.35$ | $55.82 \pm 0.56$ | $57.53 \pm 1.27$ | $61.44 \pm 2.18$ |
| ContinualGNN (M=1%) | $35.94 \pm 0.28$ | $40.06 \pm 0.85$ | $41.67 \pm 0.99$ | $41.46 \pm 1.50$ |
| ContinualGNN (M=3%) | $46.08 \pm 0.45$ | $52.71 \pm 1.22$ | $53.81 \pm 1.44$ | $55.02 \pm 1.26$ |
| ContinualGNN (M=5%) | $50.60 \pm 0.49$ | $57.22 \pm 1.22$ | $58.63 \pm 1.35$ | $60.14 \pm 1.35$ |
| GraphSAGE | $31.02 \pm 0.12$ | $31.02 \pm 0.19$ | $30.72 \pm 0.39$ | $31.92 \pm 0.46$ |
| LwF | $36.74 \pm 1.15$ | $40.65 \pm 1.06$ | $42.85 \pm 1.34$ | $48.97 \pm 1.03$ |
| ReGNN (Ours) | $\mathbf{55.02} \pm 2.01$ | $\mathbf{62.57} \pm 2.12$ | $\mathbf{64.57} \pm 2.34$ | $\mathbf{67.74} \pm 2.12$ |

graphs. Joint is the ideal upper bound situation where the memory bank contains all historical incoming graphs. **EWC** (Kirkpatrick et al., 2017) applies quadratic penalties to the model weights that are important to the previous tasks. **MAS** (Aljundi et al., 2018) utilises a regularisation term for parameters sensitive to the model performance of historical tasks. **GEM** (Lopez-Paz & Ranzato, 2017) modifies the gradients using the informative data stored in memory. **TWP** (Liu et al., 2021) preserves the topological information for previous tasks by a regularisation term. **LwF** (Li & Hoiem, 2017) distils the knowledge from the old model to the new model to keep the previous knowledge. **ER-GNN** (Zhou & Cao, 2021) samples the informative nodes from incoming graphs into the memory bank. **SSM** (Zhang et al., 2022) stores the sparsified incoming graph in the memory bank for future replay.

Although in data-free continual learning setting, no data can be stored for replay, we show that ReGNN outperforms ER-GNN (Zhou & Cao, 2021) and ContinualGNN(Wang et al., 2020), which use memory buffers for replay. Because ER-GNN (Zhou & Cao, 2021) already outperforms continual learning models EWC (Kirkpatrick et al., 2017) and GEM (Lopez-Paz & Ranzato, 2017), we only report the results for ER-GNN. More discussions about choices of baselines are available in Appendix A.

**Experimental Setup.** We use the neighborhood sampling strategy in GraphSAGE for all models when sampling is available. Details of experiment settings are available in Appendix A.

### 4.3 Main Results

Table 1 shows the main results of our model and baselines. The proposed ReGNN outperforms LwF (Li & Hoiem, 2017), which also does not require a memory buffer as our model, by a large margin. ER-GNN (Zhou & Cao, 2021) is a strong baseline modified from experience replay. Experience Replay, a typical memory based method, is a competitive baseline because it is able to access the old data during training by maintaining a memory buffer, and it often notably outperforms the regularization based methods in class-incremental learning scenario (Buzzega et al., 2020; 2021) with a small buffer size (e.g., 2% in (Buzzega et al., 2021)). The shortage of memory based methods is that they need a memory buffer to store data, which is not always feasible. We experiment with memory size M = $\{1\%, 3\%, 5\%\}$ for memory based models. Experiment results show the effectiveness of ReGNN in class-incremental setting. Our model still outperforms ER-GNN (Zhou & Cao, 2021) and ContinualGNN (Wang et al., 2020) when they have a buffer with memory size M = 5%. This is considered to be a relatively large buffer size in continual learning, given that the actual number of stored nodes, including the neighbor nodes for aggregation, is larger than our reported M.

In Tab. 5, we compare our model with more continual learning baselines on graph with larger scale. In Tab. 4, we show detailed data statistics of the benchmarks. We consistently outperform baselines under the same memory-free scenario (i.e., all previous data are dropped and are not allowed to be stored in a memory buffer).

In Tab. 6, we show results of our model and memory-based models. We show their results with (marked with 'Required' in the table) and without (marked with '-' in the table) memory buffers. We kindly remind that

Table 3: Results of replay based models on Citeseer when neighbors of the stored old nodes are not available in the memory buffer.

| Model | Default | 3-hop | 5-hop | Full |
|---|---|---|---|---|
| GraphSAGE (Hamilton et al., 2017) | 31.02 | 31.02 ($\downarrow$ **0.00**) | 30.72 ($\downarrow$ **0.30**) | 31.92 ($\uparrow$ **0.90**) |
| ContinualGNN (*) (Wang et al., 2020) | 41.86 | 45.28 ($\downarrow$ **3.42**) | 47.89 ($\downarrow$ **6.03**) | 47.59 ($\downarrow$ **5.73**) |
| ER-GNN (*) (Zhou & Cao, 2021) | 43.37 | 41.56 ($\downarrow$ **1.81**) | 40.69 ($\downarrow$ **2.68**) | 42.21 ($\downarrow$ **1.13**) |
| ER-MLP (*) (Rolnick et al., 2019) | 39.23 | 37.17 ($\downarrow$ **2.06**) | 36.32 ($\downarrow$ **2.91**) | 36.74 ($\downarrow$ **2.49**) |
| ReGNN (Ours) | 55.02 | 62.57 ($\uparrow$ **7.55**) | 64.57 ($\uparrow$ **9.55**) | 67.74 ($\uparrow$ **12.72**) |

Table 4: Dataset Statistics.

| Dataset | Nodes | Edges | Features | Classes |
|---|---|---|---|---|
| CoraFull (McCallum et al., 2000) | 19,793 | 130,622 | 8,710 | 70 |
| Arxiv (Hu et al., 2020) | 169,343 | 1,166,243 | 128 | 40 |
| Reddit (Hamilton et al., 2017) | 227,853 | 114,615,892 | 602 | 40 |
| Products (Hu et al., 2020) | 2,449,028 | 61,859,036 | 100 | 46 |

they are not our baselines because (1) we work under different scenarios (memory-based v.s. memory-free), and (2) storing previous data in a memory buffer (i.e., memory-based) makes the problem significantly easier and is not a fair comparison.

Interestingly, even without a memory buffer, our method outperforms GEM, ER-GNN, SSM on all benchmarks, and CaT on Products in Tab. 6. When evaluating the memory-based models under our memory-free scenario, where their memory buffers are removed, they lose the ability to prevent forgetting and degrade to a lower-bound performance.

Table 5: Comparison with Memory-free Baselines. CL Model refers to Continual Learning Model.

| Methods | CL Model | Memory | CoraFull | Arxiv | Reddit | Products |
|---|---|---|---|---|---|---|
| Finetuning | No | - | 1.9 | 4.3 | 4.2 | 3.6 |
| GLNN | No | - | 1.9 | 3.9 | - | 3.2 |
| NOSMOG | No | - | 1.9 | 4.3 | - | 3.6 |
| EWC | Yes | - | 2.3 | 4.0 | 4.2 | 6.1 |
| MAS | Yes | - | 1.8 | 3.9 | 8.7 | 8.2 |
| TWP | Yes | - | 16.3 | 3.1 | 7.4 | 5.2 |
| LwF | Yes | - | 1.9 | 4.3 | 4.2 | 3.6 |
| ReGNN (Ours) | Yes | - | **18.7** | **32.7** | **45.8** | **56.4** |

**Inference at Different Levels of Neighbor Information**. Table 1 shows the effectiveness of ReGNN in our practical scenario for continual graph learning illustrated in Fig. 1 and Fig. 2, where the neighborhood information for test nodes is poorly available or completely unavailable. A natural question is, whether ReGNN can also achieve good performance in different cases where the neighborhood information is provided at different levels. This could be valuable when additional information is available to help build richer connections between test nodes for more neighborhood information and better inference. We further investigate the performance of ReGNN in different cases.

Because we randomly split the data into training and test sets and also detach the test nodes from the *original graph*, we cannot control the level of neighborhood information among test nodes with the random partition process. Instead, we change the level of neighborhood information by manually linking test nodes according to their distance to each other in the *original graph*.

Table 6: Comparison with Memory-based Baselines. CL Model refers to Continual Learning Model.

| Methods | CL Model | Memory | CoraFull | Arxiv | Reddit | Products |
|---------|----------|--------|----------|-------|--------|----------|
| GEM | Yes | Required | 2.1 | 4.1 | 4.0 | 3.3 |
| | | - | 1.9 | 4.3 | 4.2 | 3.6 |
| ER-GNN | Yes | Required | 3.1 | 23.8 | 22.9 | 31.0 |
| | | - | 1.9 | 4.3 | 4.2 | 3.6 |
| SSM | Yes | Required | 12.1 | 26.9 | 39.7 | 49.2 |
| | | - | 1.9 | 4.3 | 4.2 | 3.6 |
| CaT | Yes | Required | 50.4 | 53.8 | 78.9 | 54.9 |
| | | - | 1.9 | 4.3 | 4.2 | 3.6 |
| ReGNN (Ours) | Yes | - | **18.7** | **32.7** | **45.8** | **56.4** |

Specifically, given a pair of nodes $(v, u)$ from detached test nodes, we denote the distance from $v$ to $u$ as $Dist(v, u)$, which is calculated by the number of hops from $v$ to $u$ in the *original graph*. *Original graph* refers to the complete graph before the training-test split. In this way, we can increase the neighborhood information of test nodes by manually linking pairs of test nodes. For instance, $(v, u)$ can be linked if $Dist(v, u)$ is smaller than a certain threshold $k$. After linking all node pairs $\{(v, u) | Dist(v, u) < k, v \in \mathcal{V}_{test}, u \in \mathcal{V}_{test}\}$ in the test nodes, we denote the evaluation on the test data with this additional neighborhood information as *k-hop* evaluation. Note that the way we increase the neighborhood information can involve noise if $k$ is set to a very large number, which consequently leads to a performance drop. In our experiments, we empirically choose a proper $k$ that is smaller than a threshold, as larger $k$ values fail to further improve the performance.

Experiments show that the manually added links are helpful for node classification. Because in *k-hop* evaluations, we manually select a proper $k$ and add these links according to the original complete graph. An interesting future direction is to train an additional task, for instance, link prediction, to predict these links and then link them for better node classification. In this work, we just manually add these links and focus on analyzing the ReGNN model with $k - hop$ evaluation.

Tab. 2 shows the results of this study. *Default* means no additional links are added. $k$ refers to *k-hop* evaluation, and *Full* is the evaluation method adopted by existing works in Tab. 2, i.e., linking the detached test nodes back to the graph from which they are detached for inference. Intuitively, *Full* should yield best results. Compared to the results of the *default* evaluation, increasing neighborhood information can indeed improve our model performance. ReGNN constantly outperforms other baselines under different settings, which manifests the reliability and effectiveness of our model and indicates that our model can work well in different cases with either rich or poor neighborhood information.

**Preserving Neighbor Information via Generative Replay**. Because each node is associated with a node attribute, simply replaying and using node attributes can also make predictions. Although ReGNN outperforms other baselines, we are curious to know whether generated replay data can really help ReGNN remember the neighbor information and effectively exploit them for inference, or it is just simply replaying the attribute of the input node.

To study this problem, we conduct another experiment with different baseline models **ER-GNN(*)** (Zhou & Cao, 2021), **ContinualGNN(*)** (Wang et al., 2020) and **MLP+ER** (*) (Rolnick et al., 2019), where (*) indicates the memory buffer only store nodes without their neighbors, thus the neighbor information *cannot* be replayed. Different from other models, ER-MLP does not include a GNN and uses MLP to learn from isolated node features without neighbor information. We add experiencer replay similar to ER-GNN for continual learning setting.

We start with analyzing how ER-GNN remembers the neighborhood information. ER-GNN requires a memory buffer to store old training nodes from previous tasks for replay to prevent forgetting. When replaying stored old nodes from the buffer, the ***neighbor nodes*** of the old nodes need to be sampled via the sampling strategy in graphSAGE for aggregation to compute the embeddings of stored old nodes. Thus in earlier experiments, we ***allow*** ER-GNN to store the neighbors of the stored old nodes to fulfill the model's

Table 7: Ablation study results.

| Method | Accuracy |
|---|---|
| Full Method | **55.02 ± 2.01** |
| Ablate NodeAE | 52.40 ± 2.14 |
| Use separate GraphCVAE | 51.21 ± 1.79 |
| Ablate GraphCVAE (No replay) | 31.02 ± 0.12 |

potential. In this case, the memory size of the buffet is in fact larger than our reported 1% 3% 5%, because their neighbors are also stored for the neighbor aggregation process in GraphSAGE. In this way, when ER-GNN uses old data for training, the neighborhood information of the stored old nodes is also replayed. Thus ER-GNN can remember and exploit the neighbor information during inference, which is proved by results in 2: as $k - hop$ increases, ER-GNN achieves better performance by using the richer neighbor information.

Here we experiment with another version of ER-GNN and ContinualGNN, where the neighbors of stored old nodes are **not** available, which means only their own node attributes are used for replay. 3 shows, in this case, ContinualGNN (Wang et al., 2020) still has improvements as $k$ increases because it also uses model regularization besides data replay. However, the performance of ER-GNN, which fully relies on data replay, fails to be improved as $k$ increases (i.e., richer neighborhood information is provided), which means ER-GNN cannot exploit neighborhood information anymore. This is because only isolated node attributes are replayed without neighbor aggregation thus model forgets how to exploit given neighbors. Note that only replaying node attributes helps to remember the features of isolated nodes, thus it still outperforms graphSAGE by a large margin. ContinualGNN (Wang et al., 2020) still has improvements as $k$ increases because it also uses model regularization besides data replay.

The performance of our model consistently grows with richer neighborhood information, indicating that ReGNN remembers how to exploit given neighbors for better inference by replaying our generated data. It further manifests our generated data contains helpful neighborhood information, which prevents ReGNN from forgetting it during generative replay.

### 4.4 Ablation Study

To simplify the model structure, we use the graph convolution layer in GNN as the graph encoder for the GraphCVAE. We are curious about whether sharing parameters can further improve the performance. To study this point, we train a separate GraphCVAE that owns an separate graph encoder. In this way, the GraphCVAE becomes a separate model without any shared parameters with others. It follows the representative generative replay based framework (Shin et al., 2017), where an separate generative model is trained for replay without any interactions, such as parameter sharing, with the classification module.

Through the ablation experiment, We find that sharing part of the parameters (i.e., the graph encoder) not only simplifies the model structure, but also improves the performance of the model. It indicates proper parameter sharing learns better than using two separate modules. The two reconstruction process can share some useful information with each other for better representation learning. Finally we ablate the GraphCVAE module to stop generative replay. The forgetting phenomenon becomes obvious.

A clear performance drop is observed after removing NodeAE in Tab. 7. It manifests the effectiveness of learning to reconstruct the neighbors from single node attribute embeddings with node AE module.

## 5 Conclusions and Future Work

In this paper, we find an important and practical case that has been ignored by existing works in continual graph learning: the unavailability of previous data leads to sparse neighborhood information during inference,

in additional to the challenge of catastrophic forgetting. We further propose ReGNN to jointly solve the challenges, whose effectiveness is supported by the experiments.

More advanced graph generation techniques can also be integrated into our graph encoder and decoder modules for better graph generation, which we we leave for future work. It has rarely been studied to generate large graphs with different node attributes, and most of existing works on this topic focus on molecular graph generation (Simonovsky & Komodakis, 2018; Mitton et al., 2021). Some other works (Bojchevski et al., 2018; Wang et al., 2018) learn to generate sub-graph for large graphs, which focus on the structure reconstruction. However, the node features, which are critical in many real applications, are not considered in these methods. Effective graph generation with node attributes is another under-explored topic and is beyond the scope of this paper. One promising future direction is to integrate such a generation process with the proposed method in this paper. $k - hop$ evaluations have better performance but we manually set $k$ and build links with ground truth. We will also explore automating this process so that the model can infer the connections by itself to improve model performance.

## Acknowledgement

The work is in part supported by the the U.S. Army Research Office Award under Grant Number W911NF-21-1-0109, the National Science Foundation under Grants IIS-2316305 and IIS-2316306, the Cisco Faculty Award, and the Adobe Data Science Research Award.

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

# A    Appendix

## A.1    Dataset

To evaluate the performance of ReGNN in our continual graph learning scenario, we conduct experiments on three benchmark datasets that are also used to build datasets for continual learning settings in related works (Wang et al., 2022; Liu et al., 2021; Zhou & Cao, 2021): **Cora** (Sen et al., 2008), **CiteSeer** (Sen et al., 2008), and we use **Amazon** (McAuley et al., 2015) to refer to AmazonCoBuyPhoto, a segment of Amazon co-purchasing graph. We split each dataset into several tasks, following (Liu et al., 2021; Zhou & Cao, 2021; Wang et al., 2022).

**Cora** (Sen et al., 2008) is a citation network consisting of and 5429 links and 2708 nodes classified into one of seven classes. Each node has a vector of size 1433 as its attribute.

**CiteSeer** (Sen et al., 2008) consists of 3312 publications classified into one of six classes. The citation network consists of 4732 links. Each node has a vector of size 3073 as its attribute.

**Amazon** (McAuley et al., 2015) to refer to AmazonCoBuyPhoto, a segment of Amazon co-purchasing graph. It consists of 119,043 links with 7,650 nodes of 8 classes.

To construct datasets for class-incremental graph learning, we follow (Liu et al., 2021; Zhou & Cao, 2021; Wang et al., 2022) and divide each dataset into several tasks, where each task contains several non-overlapping classes. For Cora, we divide it into three tasks. The first and second tasks consist of two classes, and the last task contains three classes. Citeseer are split into three tasks with two classes in each task. The first task of Amazon contains two classes. The second and third tasks on Amazon contain three classes.

## A.2    Experiments

### A.2.1    Experimental setup

SGD optimizer is used and the initial learning rate is set to 0.01 for Cora and Citeseer and 0.005 for Amazon. The batch size it set to 128 and run 100 epochs for each dataset. For baselines, the number of layers in GNN is set to the default value according to their papers. A standard two layer GNN is used if the information is not provided. We run each experiment five times and report the results with mean and standard deviation.

Note that otherwise specified, the buffer size M of the baseline modes (if buffer required) is the number of stored nodes *without* their neighbors. We also store their neighbors for aggregation process so that the structure information can also be replayed, which fulfills their model potential. Thus the total number of stored nodes in buffer is actually larger than M (It equals to M + the number of their neighbor nodes).

### A.2.2  Baselines

As ER-GNN is a strong baseline for graph class incremental learning, which already outperforms baselines such as EWC, GEM, we only compare with most competing baselines. Note that even ER-GNN is not necessarily our baseline because it uses memory buffers to store history data (it is a "cheat"), while we do not have any history data. For fair comparison, we should compare with models without memory buffers. Because we are the first to study graph class incremental learning without buffers, we make concessions and compare our model with competing baselines with buffers.

### A.3  Variational Autoencoders

**Autoencoders (AE).** An autoencoder is a type of a neural network, which aims to encode the input into a compressed and meaningful representation, and then decode it back such that the reconstructed input is similar as possible to the original one. The main purpose is to learn an informative representation of the data that can be used for various implications in an unsupervised manner. A typical AE learns two functions $A: \mathbb{R}^n \to \mathbb{R}^p$ and $B: \mathbb{R}^n \to \mathbb{R}^p$ that satisfy:

$$\arg \min_{A,B} \mathbb{E}[\Delta(\mathbf{x}, B \circ A(\mathbf{x})], \tag{13}$$

where $E$ is the expectation over the distribution of $x$ and $\Delta$ is the reconstruction loss function, which measures the distance between the output of the decoder and the input.

**Variational Autoencoders (VAE).** (Kingma & Welling, 2013) VAE is a generative model that is similar to AE in structure. It provides a formulation in which the encoding $z$ is interpreted as a latent variable in a probabilistic generative model. And a probabilistic decoder is defined by a likelihood function $p_\theta(\mathbf{x}, \mathbf{z})$ parameterized by $\theta$. Alongside a prior distribution $p_\theta(\mathbf{z})$ over the latent variables, the posterior distribution $p_\theta(\mathbf{z}|\mathbf{x}) \propto p(\mathbf{z})p_\theta(\mathbf{x}|\mathbf{z})$ can then be interpreted as a probabilistic encoder.

To avoid the huge complexity, the approach simultaneously learns both the parameters of $p_\theta(\mathbf{x}|\mathbf{z})$ as well as those of a posterior approximation $q_\phi(\mathbf{z}|\mathbf{x})$. This is achieved by maximizing the evidence lower bound (ELBO):

$$\mathcal{L}(\phi, \theta; \mathbf{x}) = \mathbb{E}_{q_\phi(\mathbf{z}|\mathbf{x})} \left[\log p_\theta(\mathbf{x}, \mathbf{z}) - \log q_\phi(\mathbf{z}|\mathbf{x})\right], \tag{14}$$

with $\mathcal{L}(\phi, \theta; \mathbf{x}) \leq \log p_\theta(\mathbf{x})$. Wide flexibility in choice of encoder and decoder models is allowed because the ELBO can be maximized via gradient descent as long as $p_\theta(\mathbf{x}|\mathbf{z})$ and $q_\theta(\mathbf{z}|\mathbf{x})$ can be computed point wise, and are differentiable with respect to their parameters. Conditional VAE (CVAE) is slightly different from VAE in that, the encoder and decoder are conditioned on $\mathbf{x}$ and another given variable $\mathbf{c}$. In our case, it is the class label of $\mathbf{x}$.

### A.4  Continual Graph Learning

In this section, we disucss the motivation of our setting.

We first illustrate (1) the differences of our motivation and previous works, then explain (2) why we also evaluate on two citation graphs, in addition to the AmazonCoBuying network, which is a real-world case for our motivation.

In short, our proposed scenario strictly follows continual learning, where training data are dropped once used (thus not available at both training and inference stage) due to multiple reasons like GDPR data regulation, privacy issues.

This motivation is very different from existing works such as ContinualGNN (Wang et al., 2020), where training nodes are available during both training and inference stage, and they simply choose to avoid using

previous training data at current training stage for efficiency. In our case, previous training data are unavailable due to multiple reasons mentioned above, which strictly follows the continual learning scenario (Van de Ven & Tolias, 2019). Note that although existing works mentioned "continual learning" (in a general sense) in their presentation, in fact it is not a standard continual learning setting because their history data are actually not unavailable. That is why we propose our scenario, where history data are actually unavailable due to practical reasons.

We do not use citation networks as real-world examples, instead, we use them to build class-incremental tasks as an evaluation of our model, which are also adopted in related works (Liu et al., 2021; Zhou & Cao, 2021). We just follow them to evaluate our model performance.

Besides, we also experiment with the AmazonCoBuying network, where it is natural that history nodes are not available after use due to storage constraints, privacy issues (for example, transaction records have to be deleted due to data regulation or user preference), etc. In these cases, training data have to be dropped after training and not available across future training and test stages.

In the following, We summarize the motivation of our practical scenario from three aspects.

**(1) Comparison with Existing Works**

Let us start from continual learning (CL), in typical CL, previous training data are not available due to a series of reasons, such as privacy issues, deletion by users, storage pressure, etc. Graph continual learning, is a type of continual learning (CL) but on graph. Therefore, existing works follow CL and assume that previous training data are already dropped and not available during training. However, the uniqueness of graph learning is, the training data could be used during inference.

Here is the problem: existing typical continual learning works often focus on image classification, thus dropping previous training data does not influence the inference (because in image classification, training data is not required during inference). However, in graph learning, training data is often required during inference. But according to the standard setting in continual learning, training data is already dropped and no longer available.

However, existing graph continual learning works simply ignored the fact that "training data is already dropped and no longer available" and still use them for inference, which is contradictory. Our work corrects this contradiction.

**(2) Practical Scenarios**

In practice, frequently, historical data is unavailable due to privacy issue, deletion by users, storage pressure, etc. For instance, in social networks or user interaction graphs, history records (nodes and links) will be deleted and thus become unavailable due to customer data storage consent, such as the 30-days right-of-erasure in General Data Protection Regulation (GDPR). Besides, they can also be unavailable if users choose to hide or delete their history records.

Besides citation networks, we have also used the Amazon Co-purchasing networks, which could be more practical because historical data could be hidden or deleted by users, and they could also become unavailable due to customer data storage consent (such as 30-days right-of-erasure in GDPR). In this case, previous training data is no longer available no matter for current training or testing.

**(3) Citation Benchmarks as Simulator**

In public citation networks such as Cora, existing papers are always available for inference. In fact they are even always available for training as well. Although in public citation networks, papers are in fact always available for both training and inference, existing works still assume that previous papers are not available for training because they just want to use citation networks to simulate the cases where the

previous data is unavailable. Citation networks are commonly used because they are prevalent.

We also use citation networks for the same purpose: to simulate the cases where previous data is dropped and unavailable due to privacy issues, deletion by users, storage pressure, etc.

