# OpenReview forum: "Revealing an Overlooked Challenge in Class-Incremental Graph Learning"
_TMLR — Accepted by TMLR_

### Review · Reviewer_Cw9c · 2024-02-01

**Summary Of Contributions:**

The authors study the graph continual learning problem, where they identify the problem that prior works assume the past graph data is available at inference. The authors propose ReplayGNN as a solution for this challenge.

**Audience:**

Yes

**Claims And Evidence:**

Yes

**Requested Changes:**

- Compare with distillation-based methods [1,2]. If they are not applicable, please explain the reason in detail.

- Test with more benchmark datasets, such as those in Open Graph Benchmark or datasets that can be found in the PyTorch geometric library. If they are not applicable, please explain the reason.

**Strengths And Weaknesses:**

## Strengths

- Interesting problem. It is critical to study the graph continual learning without accessing the past data at the inference stage.

## Weaknesses

- The paper should be better polished with improved clarity.

- The experiment is not extensive. Not only some reasonable baselines are not compared, but also the used graph datasets are rather simple.

While I like the problem that the authors study, I think several weaknesses can be improved for the current manuscript. First, the paper should be better polished with improved clarity. For instance, the problem formulation is not clear to me, especially the description after performing $T^m$ tasks. Do the authors mean there are test nodes from each $D^t$ and they want to predict their label at inference?

Regarding the experiment, while the authors demonstrate some positive results compared to the tested baselines, it is only for rather simple graph datasets such as Cora. I wonder if this is common for graph continual learning and if more (challenging) benchmark datasets can be used. Also, according to the current setting, I feel it is natural to compare with the distillation-based methods (i.e., MLP distill from GNNs) [1,2]. Is there any reason why this type of method is not included as baselines? I think it is critical to compare these approaches in order to verify that constructing a "graph" for test nodes and then applying GNNs is reasonable (i.e., the proposed ReplayGNN). Based on the current experiment, it is hard for me to advocate for this work.

##References:

[1] Graph-less neural networks: Teaching old mlps new tricks via distillation, Zhang et al. ICLR 2022.

[2] Learning MLPs on Graphs: A Unified View of Effectiveness, Robustness, and Efficiency, Tian et al. ICLR 2023.

---

> ### Author Response · Authors · 2024-04-02
> **Response to Reviewer Cw9c (Part 1)**
>
> ### Question 1
>
> We apologize for the confusion, and your understanding is correct!
>
> ### Question 2 (Weakness 1 and Weakness 2)
>
> (1) Yes, we adopt the commonly used benchmarks in graph continual learning [2,3,4,5,6], such as such as Cora and Citeseer.
>
> (2) Thanks for sharing the two interesting works [10,11]! We did not consider the two works because they are not continual graph learning methods. Instead, they are backbone graph learning models such as GraphSAGE, GAT or GCN. If we directly evaluate them [10,11] in our continual learning benchmarks, they would suffer from significant performance drop as they are not designed for continual learning therefore cannot remember learner knowledge from previous tasks.
>
> In continual graph learning, continual graph learning methods are often built upon a backbone graph learning model. We choose graphSAGE because it is representative and more commonly used in continual graph learning [3,9].
>
> To adapt GLNN [10] and NOSMOG [11] into a continual graph learning model, new designs to prevent catastrophic forgetting in continual learning are required for them, separately. This would be a new research topic beyond the scope of this paper, because we need to design new continual learning modules tailored for the two new backbones.
>
> Besides, when comparing with existing baselines, it is fairer that we all use the same backbone graph learning model. For instance, when comparing with [3,9], which use GraphSASE as backbone model, we can better validate the effectiveness of our generative replay method in preventing forgetting if we also use  GraphSASE as backbone as well. If we design a new continual graph learning model with a more powerful backbone, say, NOSMOG, we might wonder if our improvements over the baselines come from our better generative replay (less forgetting) or the better performance of NOSMOG itself.
>
> Our aim is to develop a more effective generative replay method for continual graph learning, rather than focusing on improving the backbone model for general graph learning. The objective of GLNN and NOSMOG, however, is the latter.
>
> Although GLNN [10] and NOSMOG [11] are not continual learning models, we still compare our model with them as you requested in Table 2. An obvious performance gap is observed. This is reasonable as they have not designed modules to prevent the catastrophic forgetting problem.
>
> We also compare our model with more baselines on large scale graphs (Table 1) as you suggested. We consistently outperform baselines under the same memory-free scenario (i.e., all previous data are dropped and are not allowed to be stored in a memory buffer) in Table 2.
>
> In Table 3, we show results of our model and memory-based models. We show their results with (marked with 'Required' in the table) and without (marked with '-' in the table) memory buffers. **We kindly note that they are not our baselines because (1) we work under different scenarios (memory-based v.s. memory-free), and (2) storing previous data in a memory buffer (i.e., memory-based) makes the problem significantly easier and is not a fair comparison.**
>
> Interestingly, even without a memory buffer, our method outperforms GEM, ER-GNN, SSM on all benchmarks, and CaT on Products. When evaluating the memory-based models under our memory-free scenario, where their memory buffers are removed, they lose the ability to prevent forgetting and degrade to a lower-bound performance.

---

> ### Author Response · Authors · 2024-04-02
> **Response to Reviewer Cw9c (Part 2)**
>
> Table.1: Datasets
>
> | Dataset  |      | Nodes     |      | Edges       |      | Features |      | Classes |
> | -------- | ---- | --------- | ---- | ----------- | ---- | -------- | ---- | ------- |
> | CoraFull |      | 19,793    |      | 130,622     |      | 8,710    |      | 70      |
> | Arxiv    |      | 169,343   |      | 1,166,243   |      | 128      |      | 40      |
> | Reddit   |      | 227,853   |      | 114,615,892 |      | 602      |      | 40      |
> | Products |      | 2,449,028 |      | 61,859,036  |      | 100      |      | 46      |
>
>
>
> Table. 2: Comparison with memory-free baselines
>
> | Methods     |      | Continual Learning Model |      | Memory |      | CoraFull |      | Arxiv    |      | Reddit   |      | Products |
> | ----------- | ---- | ------------------------ | ---- | ------ | ---- | -------- | ---- | -------- | ---- | -------- | ---- | -------- |
> | Finetuning  |      | No                       |      | -      |      | 1.9      |      | 4.3      |      | 4.2      |      | 3.6      |
> | GLNN [10]   |      | No                       |      | -      |      | 1.9      |      | 3.9      |      | -        |      | 3.2      |
> | NOSMOG [11] |      | No                       |      | -      |      | 1.9      |      | 4.3      |      | -        |      | 3.6      |
> | EWC [12]    |      | Yes                      |      | -      |      | 2.3      |      | 4.0      |      | 4.2      |      | 6.1      |
> | MAS [13]    |      | Yes                      |      | -      |      | 1.8      |      | 3.9      |      | 8.7      |      | 8.2      |
> | TWP [14]    |      | Yes                      |      | -      |      | 16.3     |      | 3.1      |      | 7.4      |      | 5.2      |
> | LwF [15]    |      | Yes                      |      | -      |      | 1.9      |      | 4.3      |      | 4.2      |      | 3.6      |
> | **Ours**    |      | Yes                      |      | -      |      | **18.7** |      | **32.7** |      | **45.8** |      | **56.4** |
>
>
>
> Table. 3: Comparison with memory-based baselines
>
> | Methods    |      | Continual Learning Model |      | Memory   |      | CoraFull |      | Arxiv    |      | Reddit   |      | Products |
> | ---------- | ---- | ------------------------ | ---- | -------- | ---- | -------- | ---- | -------- | ---- | -------- | ---- | -------- |
> | GEM [16]   |      | Yes                      |      | Required |      | 2.1      |      | 4.1      |      | 4.0      |      | 3.3      |
> |            |      |                          |      | -        |      | 1.9      |      | 4.3      |      | 4.2      |      | 3.6      |
> | ER-GNN [3] |      | Yes                      |      | Required |      | 3.1      |      | 23.8     |      | 22.9     |      | 31.0     |
> |            |      |                          |      | -        |      | 1.9      |      | 4.3      |      | 4.2      |      | 3.6      |
> | SSM [17]   |      | Yes                      |      | Required |      | 12.1     |      | 26.9     |      | 39.7     |      | 49.2     |
> |            |      |                          |      | -        |      | 1.9      |      | 4.3      |      | 4.2      |      | 3.6      |
> | CaT [6]    |      | Yes                      |      | Required |      | 50.4     |      | 53.8     |      | 78.9     |      | 54.9     |
> |            |      |                          |      | -        |      | 1.9      |      | 4.3      |      | 4.2      |      | 3.6      |
> | **Ours**   |      | Yes                      |      | -        |      | **18.7** |      | **32.7** |      | **45.8** |      | **56.4** |

---

> ### Author Response · Authors · 2024-04-02
> **Response to Reviewer Cw9c (Part 3)**
>
> ### Reference
>
> [1] Hamilton and Ying et al., Inductive Representation Learning on Large Graphs, NeurIPS 2017
>
> [2] Huihui Liu, Yiding Yang, and Xinchao Wang. Overcoming catastrophic forgetting in graph neural networks. AAAI, 2021
>
> [3] Fan Zhou and Chengtai Cao. Overcoming catastrophic forgetting in graph neural networks with experience replay. AAAI, 2021
>
> [4] Chen Wang, Yuheng Qiu, Dasong Gao, and Sebastian Scherer. Lifelong graph learning. CVPR 2022
>
> [5] Zhang et al., CGLB: Benchmark Tasks for Continual Graph Learning, NeurIPS 2022
>
> [6] Liu et al., CaT: Balanced Continual Graph Learning with Graph Condensation, ICDM 2023
>
> [7] Wang et al., A Comprehensive Survey of Continual Learning: Theory, Method and Application, TPAMI 2024
>
> [8] [General Data Protection Regulation](https://en.wikipedia.org/wiki/General_Data_Protection_Regulation)
>
> [9] Wang et al., Streaming Graph Neural Networks via Continual Learning, CIKM 2020
>
> [10] Graph-less neural networks: Teaching old mlps new tricks via distillation, Zhang et al. ICLR 2022.
>
> [11] Learning MLPs on Graphs: A Unified View of Effectiveness, Robustness, and Efficiency, Tian et al. ICLR 2023.
>
> [12] J. Kirkpatrick et al., “Overcoming catastrophic forgetting in neural networks,” CoRR, vol. abs/1612.00796, 2016.
>
> [13] R. Aljundi et al., “Memory aware synapses: Learning what (not) to forget,” in ECCV, 2018.
>
> [14] H. Liu et al., “Overcoming catastrophic forgetting in graph neural networks,” in AAAI, 2021
>
> [15] Z. Li and D. Hoiem, “Learning without forgetting,” TPAMI, 2018
>
> [16] D. Lopez-Paz and M. Ranzato, “Gradient episodic memory for continual learning,” in NeurIPS, 2017
>
> [17] Sparsified subgraph memory for continual graph representation learning,” in ICDM, 2022

---

> > ### Comment · Reviewer_Cw9c · 2024-04-02
> > **Thanks for the additional experiments and clarification**
> >
> > I really appreciate the effort. I am now satisfied with the new experiments with new baselines and larger datasets, and thus I will change my "claim and evidence" from No to Yes.
> >
> > If I understand correctly, the authors show that the distillation-based method (MLP distill GNN) does not work in the continual learning setting. Originally I thought this class of methods could "store" information from tasks $T^t$ to MLP in an online fashion, but it seems that I am wrong. I wonder how do the authors run GLNN or NOSMOG? My original thought was just to train a GNN using only $T^t$ and update the MLP student accordingly in an online fashion. That is, learn/finetune a $GNN_t$ from $T^t$, and then finetune the student MLP using $GNN_t$ and $T^t$. In this case, it seems that at least the student MLP has the chance to "learn" the information from the entire graph (i.e., the union of $T^t$). However, it is also possible that the dataset $T^t$ is not large enough to make $GNN_t$ trained well.
> >
> > Nevertheless, it is unreasonable to ask the authors to modify GLNN or NOSMOG along with this idea, as it would be another paper. Still, I think the part of constructing a "test graph" feels weird and unnatural to me. I encourage the authors to think if a distillation-based method combined with continual learning can be a better solution. It is also possible that I missed something or I'm completely wrong. In this case, I would appreciate the authors to point them out.

---

> ### Author Response · Authors · 2024-04-02
> **Thanks for your recognition**
>
> Thanks for your timely response and recognition!
>
> **1. Distillation-based method**
>
> Let's first recap the standard class-incremental continual graph learning setting: we assume that the graph data comes in a streaming manner, and we sequentially learn $N$ tasks one by one.
>
> We follow the standard pipeline when adapting GLNN/NOSMOG into this setting: in each task, we first train the GNN teather and then distill to the MLP student, with training data for current task.
>
> We try to follow your understanding (please feel free to correct us if we misunderstand your idea): you seem to believe that GLNN/NOSMOG should work because the MLP student has learnt from the GNN teacher for all $N$ tasks. While the MLP student indeed learns from the GNN teacher in each task, the GNN teacher itself suffers from forgetting: it sequentially learns $N $ tasks, same to the Finetuning baseline.
>
> *It means the GNN teacher is not regularized by any methods to prevent forgetting learnt tasks.*
>
> When the distillation is performed with the GNN teacher and MLP student, the student also quickly forgets previous knowledge as "no one" reminds it of learnt knowledge.
>
> Interestingly, one intuitive idea is that, we might store previous GNN teachers, and during the current task, we distill the knowledge in both the previous GNN teacher and the current GNN teacher to the MLP student, with current training data. In fact, it is the idea of LwF [15], one of our baselines, where we distill the knowledge from the previous GNN to the current GNN (with current training data), while we learn from the current task.
> It sounds reasonable, but the key problem here is the data for distillation: a good distillation from old network to current network requires **old data**, which is widely observed in continual learning research. However, we only have current data. That is why distillation-based continual learning methods like LwF [15] is often not effective in a memory-free scenario.
>
> Although current distillation-based continual learning methods suffer from the above problem in a memory-free scenario, we agree with you that, it is interesting and presents a good potential opportunity for future research. The major challenge is to find (or even generate) the "appropriate data" that better matches the old data distribution for effective distillation.
>
> **2. Constructing a "test graph"**
>
> We would like to clarify that, constructing a "test graph" for k-hop evaluation is not necessary in our study or for our model.
> In our tables for main results (Tab. 1 in paper, and additional tables in our rebuttal), the reported results are not under the k-hop evaluation, instead, we directly evaluate models with incoming test nodes without any additional k-hop links.
>
> We conduct k-hop evaluation just for analytic experiments. The simple and intuitive assumption behind it is: "nearby nodes have similar representations" [1]. Actually, when k=1 (i.e., 1-hop), it is the default setting: if there is a link (1-hop) between two test nodes, just let it be. Naturally, we are curious that, what if we link test nodes with 2-hop, 3-hop, ..., k-hop distance? After all, they also share similar representations to different extends because thay are close to each other. It is an exploratory study and not necessary for our model and our setting.

---

### Review · Reviewer_pB6o · 2024-02-05

**Summary Of Contributions:**

This paper focus on a scenario, in which the testing nodes are accompanied with very sparse edge connections, which is inconsistent with the training phase with full neighborhood information.

Besides the novel setting, the proposed method contains one graph encoder, one node encoder, one classifier, and one graph decoder. The encoder encode the node and structure information into laten space. The decoders serve to guide the model to encode the node information in the latent representations, and provides structural information during testing, when the graph structure is sparse.

**Audience:**

Yes

**Broader Impact Concerns:**

The studied problem should not have significant ethical concerns.

**Claims And Evidence:**

No

**Requested Changes:**

1. Justification of the practical importance of the proposed setting.

2. More thorough discussion on the related works.

**Strengths And Weaknesses:**

Strengths:

1. The paper proposes a novel setting, with a reasonable model to tackle this new setting.


Weaknesses

1. The proposed setting assumes that the streaming testing nodes are typically isolated from the previously observed training graph, which is seemingly reasonable, but lacks real-world correspondence. For example, the citation networks used in this paper (cora and citeseer), when new papers appear, why couldn't they be connected to existing papers with citation relationship? It seems that inserting the new paper into the existing citation network does not incur much computational cost, since the citation information are already included in the paper's data (e.g. biblography). Therefore, it is necessary to justify that the so called 'overlooked problem' is indeed practically important.

2. The baselines are a bit old. Actually there is only one method designed for continual learning on graphs included in this paper (ER-GNN). This paper is a branch of to continual learning on graph data, which has multiple publications in 2022 and 2023. The authors are encouraged to compare or discuss the recent literature.

---

> ### Author Response · Authors · 2024-04-02
> **Response to Reviewer pB6o (Part 1)**
>
> ### Question 1
>
> *"For example, the citation networks used in this paper (cora and citeseer), when new papers appear, why couldn't they be connected to existing papers with citation relationship?"*
>
> We would like to address your concern from three aspects:
>
> (1) Comparison with Existing Works
>
> (2) Practical Scenario in Amazon Data
>
> (3) Citation Benchmarks as Simulator
>
> **(1) Comparison with Existing Works**
>
> Let us start from continual learning (CL) [7], in typical CL, previous training data are not available due to a series of reasons, such as privacy issues [8], deletion by users, storage pressure, etc. Graph continual learning [2,3,4,5], is a type of continual learning (CL) but on graph. Therefore, existing works follow CL and assume that *previous training data are already dropped and not available during training.* However, the uniqueness of graph learning is, the training data could be used during inference.
>
> Here is the problem: existing typical continual learning works often focus on image classification, thus dropping previous training data does not influence the inference (because in image classification, training data is not required during inference). However, in graph learning, training data is often required during inference. But according to the standard setting in continual learning, training data is already dropped and no longer available.
>
> However, existing graph continual learning works [2,3,4,5,6,17] simply ignored the fact that "training data is already dropped and no longer available" and still use them for inference, which is contradictory. Our work corrects this contradiction.
>
> **(2) Practical Scenarios**
>
> We would like to clarify that, in practice, frequently, historical data is unavailable due to privacy issue [8], deletion by users, storage pressure, etc. For instance, in social networks or user interaction graphs (Fig. 1 in the paper), history records (nodes and links) will be deleted and thus become unavailable due to customer data storage consent, such as the 30-days right-of-erasure in General Data Protection Regulation (GDPR). Besides, they can also be unavailable if users choose to hide or delete their history records.
>
> Besides citation networks, we have also used the Amazon Co-purchasing networks, which could be more practical because historical data could be hidden or deleted by users, and they could also become unavailable due to customer data storage consent (such as 30-days right-of-erasure in GDPR). In this case, previous training data is no longer available no matter for current training or testing.
>
> **(3) Citation Benchmarks as Simulator**
>
> We agree that, in public citation networks such as Google Scholar, existing papers are always available for inference. In fact they are even always available for training as well: they are public and will not disappear. Therefore, continual learning is not designed for this type of public data, which are free from the unavailability assumption in continual learning due to privacy issue [8], deletion by users, storage pressure, etc.
>
> Although in public citation networks, papers are in fact always available for both training and inference, existing works [2,3,4,5,6,17] still assume that previous papers are not available for training because they just want to use citation networks to **simulate** the cases where the previous data is unavailable. Citation networks are commonly used because they are prevalent.
>
> We also use citation networks for the same purpose: to simulate the cases where previous data is dropped and unavailable due to privacy issues [8], deletion by users, storage pressure, etc.

---

> ### Author Response · Authors · 2024-04-02
> **Response to Reviewer pB6o (Part 2)**
>
> ### Question 2
>
> Thanks for your advice. In our revised paper (Appendix), we have added more baselines from more recent papers.
>
> We compare our model with more baselines on large scale graphs (Table 1) as you suggested. Our method consistently outperforms baselines under the same memory-free scenario (i.e., all previous data are dropped and are not allowed to be stored in a memory buffer) in Table 2.
>
> In Table 3, we show results of our model and memory-based models. We show their results with (marked with 'Required' in the table) and without (marked with '-' in the table) memory buffers. **We kindly note that they are not our baselines because (1) we work under different scenarios (memory-based v.s. memory-free), and (2) storing previous data in a memory buffer (i.e., memory-based) makes the problem significantly easier and is not a fair comparison.**
>
> Interestingly, even without a memory buffer, our method outperforms GEM, ER-GNN, and SSM on all benchmarks, and CaT on Products. When evaluating the memory-based models under our memory-free scenario, where their memory buffers are removed, they lose the ability to prevent forgetting and degrade to a lower-bound performance.
>
> Table.1: Datasets
>
> | Dataset  |      | Nodes     |      | Edges       |      | Features |      | Classes |
> | -------- | ---- | --------- | ---- | ----------- | ---- | -------- | ---- | ------- |
> | CoraFull |      | 19,793    |      | 130,622     |      | 8,710    |      | 70      |
> | Arxiv    |      | 169,343   |      | 1,166,243   |      | 128      |      | 40      |
> | Reddit   |      | 227,853   |      | 114,615,892 |      | 602      |      | 40      |
> | Products |      | 2,449,028 |      | 61,859,036  |      | 100      |      | 46      |
>
>
>
> Table. 2: Comparison with memory-free baselines
>
> | Methods     |      | Continual Learning Model |      | Memory |      | CoraFull |      | Arxiv    |      | Reddit   |      | Products |
> | ----------- | ---- | ------------------------ | ---- | ------ | ---- | -------- | ---- | -------- | ---- | -------- | ---- | -------- |
> | Finetuning  |      | No                       |      | -      |      | 1.9      |      | 4.3      |      | 4.2      |      | 3.6      |
> | GLNN [10]   |      | No                       |      | -      |      | 1.9      |      | 3.9      |      | -        |      | 3.2      |
> | NOSMOG [11] |      | No                       |      | -      |      | 1.9      |      | 4.3      |      | -        |      | 3.6      |
> | EWC [12]    |      | Yes                      |      | -      |      | 2.3      |      | 4.0      |      | 4.2      |      | 6.1      |
> | MAS [13]    |      | Yes                      |      | -      |      | 1.8      |      | 3.9      |      | 8.7      |      | 8.2      |
> | TWP [14]    |      | Yes                      |      | -      |      | 16.3     |      | 3.1      |      | 7.4      |      | 5.2      |
> | LwF [15]    |      | Yes                      |      | -      |      | 1.9      |      | 4.3      |      | 4.2      |      | 3.6      |
> | **Ours**    |      | Yes                      |      | -      |      | **18.7** |      | **32.7** |      | **45.8** |      | **56.4** |
>
>
>
> Table. 3: Comparison with memory-based baselines
>
> | Methods    |      | Continual Learning Model |      | Memory   |      | CoraFull |      | Arxiv    |      | Reddit   |      | Products |
> | ---------- | ---- | ------------------------ | ---- | -------- | ---- | -------- | ---- | -------- | ---- | -------- | ---- | -------- |
> | GEM [16]   |      | Yes                      |      | Required |      | 2.1      |      | 4.1      |      | 4.0      |      | 3.3      |
> |            |      |                          |      | -        |      | 1.9      |      | 4.3      |      | 4.2      |      | 3.6      |
> | ER-GNN [3] |      | Yes                      |      | Required |      | 3.1      |      | 23.8     |      | 22.9     |      | 31.0     |
> |            |      |                          |      | -        |      | 1.9      |      | 4.3      |      | 4.2      |      | 3.6      |
> | SSM [17]   |      | Yes                      |      | Required |      | 12.1     |      | 26.9     |      | 39.7     |      | 49.2     |
> |            |      |                          |      | -        |      | 1.9      |      | 4.3      |      | 4.2      |      | 3.6      |
> | CaT [6]    |      | Yes                      |      | Required |      | 50.4     |      | 53.8     |      | 78.9     |      | 54.9     |
> |            |      |                          |      | -        |      | 1.9      |      | 4.3      |      | 4.2      |      | 3.6      |
> | **Ours**   |      | Yes                      |      | -        |      | **18.7** |      | **32.7** |      | **45.8** |      | **56.4** |

---

> ### Author Response · Authors · 2024-04-02
> **Response to Reviewer pB6o (Part 3)**
>
> ### Reference
>
> [1] Hamilton and Ying et al., Inductive Representation Learning on Large Graphs, NeurIPS 2017
>
> [2] Huihui Liu, Yiding Yang, and Xinchao Wang. Overcoming catastrophic forgetting in graph neural networks. AAAI, 2021
>
> [3] Fan Zhou and Chengtai Cao. Overcoming catastrophic forgetting in graph neural networks with experience replay. AAAI, 2021
>
> [4] Chen Wang, Yuheng Qiu, Dasong Gao, and Sebastian Scherer. Lifelong graph learning. CVPR 2022
>
> [5] Zhang et al., CGLB: Benchmark Tasks for Continual Graph Learning, NeurIPS 2022
>
> [6] Liu et al., CaT: Balanced Continual Graph Learning with Graph Condensation, ICDM 2023
>
> [7] Wang et al., A Comprehensive Survey of Continual Learning: Theory, Method and Application, TPAMI 2024
>
> [8] [General Data Protection Regulation](https://en.wikipedia.org/wiki/General_Data_Protection_Regulation)
>
> [9] Wang et al., Streaming Graph Neural Networks via Continual Learning, CIKM 2020
>
> [10] Graph-less neural networks: Teaching old mlps new tricks via distillation, Zhang et al. ICLR 2022.
>
> [11] Learning MLPs on Graphs: A Unified View of Effectiveness, Robustness, and Efficiency, Tian et al. ICLR 2023.
>
> [12] J. Kirkpatrick et al., “Overcoming catastrophic forgetting in neural networks,” CoRR, vol. abs/1612.00796, 2016.
>
> [13] R. Aljundi et al., “Memory aware synapses: Learning what (not) to forget,” in ECCV, 2018.
>
> [14] H. Liu et al., “Overcoming catastrophic forgetting in graph neural networks,” in AAAI, 2021
>
> [15] Z. Li and D. Hoiem, “Learning without forgetting,” TPAMI, 2018
>
> [16] D. Lopez-Paz and M. Ranzato, “Gradient episodic memory for continual learning,” in NeurIPS, 2017
>
> [17] Sparsified subgraph memory for continual graph representation learning,” in ICDM, 2022

---

> ### Comment · Reviewer_pB6o · 2024-04-04
> **Thanks for the detailed response from the authors**
>
> Thanks for the detailed response from the authors. I'm overall satisfied with the response, and my previous concerns on 1. practical scenarios, and 2. involving more related methods, are resolved. The paper would be improved by incorporating these additional components.
>
> By the way, one statement in the rebuttal may be improper: 'they are public and will not disappear. Therefore, continual learning is not designed for this type of public data'. As far as I'm concerned, even if all the previous data are always available, we still need continual learning, because the model still has to constantly adapt to new data, and the forgetting problem still exists. When the previous data are in large amounts, we cannot always retrain the model over all data, so we still need contiual learning techniques to handle the incremental learning on new data.

---

> > ### Author Response · Authors · 2024-04-04
> > **Thanks for your recognition and feedback**
> >
> > Thanks for your recognition to our response!
> >
> > We agree and appreciate your helpful comment: alleviating the training cost is also a practical and useful case of continual learning. We have added the discussion from this aspect in our revised paper (Introduction Section):
> >
> > *"Furthermore, even in scenarios where all prior data remains accessible, continual graph learning still holds significance. This is because it is advantageous for the model to incrementally learn from new data, rather than undergoing a complete retraining process from scratch."*
> >
> > Thanks again for your timely and helpful advice. We would like to address them if you have any further concerns.

---

### Review · Reviewer_QCHX · 2024-03-20

**Summary Of Contributions:**

The paper studies class-incremental graph learning under (what they argue to be) a more realistic assumption: historical data may be unavailable during both training and inference. This poses additional challenges for graph learning, e.g. stemming from (neighborhood) sparsity. The authors propose a generative replay based model that does not (explicitly) rely on memory buffers. Their model has to key modules: a conditional (on class labels) VAE to generate old data for replay, and NodeAE that directly maps node attributes. The two modules share a shared graph decode.

They main contribution is highlighting the more realistic learning setting and proposing a model that tackles the challenges that come with it.

**Audience:**

Yes

**Broader Impact Concerns:**

None.

**Claims And Evidence:**

Yes

**Requested Changes:**

None.

**Strengths And Weaknesses:**

Strengths:
- The proposed model is well-motivated (and simple which is bonus) to handle the challenges stemming from the more realistic learning scenario.
- ReGNN outperforms the baselines with a good margin, includes baselines that additionally use a memory buffer
- The experimental evaluation is thorough and convincing (although I'm not very familiar with the graph CL literature).

Weaknesses:
-  Manually linking test nodes based on distance in the original graph seems like an odd choice. A different way to control the level of neighborhood information (e.g. a more "stratified" train-test split) might be more informative. However, I believe this can be left for future work.
- The datasets are not a good representative of real-world graph CL problems (even though they are often used as benchmarks). For example, citation networks are not likely to "evolve" in a class-incremental manner. This may lead to overly optimistic methods for all baselines.

---

> ### Author Response · Authors · 2024-04-02
> **Response to Reviewer QCHX**
>
> ## R1
>
> ### Question 1
>
> Our current method is based on a simple and intuitive assumption that "nearby nodes have similar representations" [1]. We agree that there would be better ways to control the level of neighborhood information, such as creating a more "stratified" train-test split as you mentioned. However, one interesting thing we would like to explore in the paper is, under a random train-test split (within classes), whether linking nodes which share similar representations (i.e., closer to each other under the assumption) during inference can help improve the test accuracy. This is practical because in real-world applications, within a given task, train-test data is more likely to be randomly split.
>
> In future work, we would like to study how to better control the level of neighborhood information following your advice. Thanks for your helpful suggestion!
>
> ### Question 2
>
> We adopt the benchmarks as they are commonly used in graph continual learning [2,3,4,5,6]. We agree that existing benchmarks are not perfect and we also look forward to new benchmarks in this field. As our work focuses more on the methodology instead of creating new benchmarks, we follow existing works and evaluate our model on the benchmarks.
>
> Although existing benchmarks are not perfect, they actually can reflect graph continual learning (**class-incremental setting**) in real-world applications. For instance, in citation networks, how the graph are evolved depends on from which perspective we study: if we fix the number of research topic (or area), then it is a **data-incremental setting**, where new papers from all existing topics (e.g., CV, NLP, etc.) are incrementally added. If we consider the scenario where new research topics (or area) are emerging, then it is a **class-incremental setting**, which researches on graph continual learning [2,3,4] are particularly interested. This setting is also practical because new research areas are always emerging. For instance, "graph continual learning" itself is a new research topic that has emerged in recent years. Besides, we also use amazon co-purchasing networks, where new categories are incrementally added.
>
> In the revised paper (Appendix), we have added new experiments.Specifically, we have compared our model with more baselines on more large scale graphs (Table 1). We include Reddit and Products, in addition to the citation graphs.
>
> New results are shown in Table 2. Our method consistently outperforms baselines under the same memory-free scenario (i.e., all previous data are dropped and are not allowed to be stored in a memory buffer).
>
> Table.1: Datasets
>
> | Dataset  |      | Nodes     |      | Edges       |      | Features |      | Classes |
> | -------- | ---- | --------- | ---- | ----------- | ---- | -------- | ---- | ------- |
> | CoraFull |      | 19,793    |      | 130,622     |      | 8,710    |      | 70      |
> | Arxiv    |      | 169,343   |      | 1,166,243   |      | 128      |      | 40      |
> | Reddit   |      | 227,853   |      | 114,615,892 |      | 602      |      | 40      |
> | Products |      | 2,449,028 |      | 61,859,036  |      | 100      |      | 46      |
>
> Table. 2: Comparison with memory-free baselines
>
> | Methods    |      | Continual Learning Model |      | Memory |      | CoraFull |      | Arxiv    |      | Reddit   |      | Products |
> | ---------- | ---- | ------------------------ | ---- | ------ | ---- | -------- | ---- | -------- | ---- | -------- | ---- | -------- |
> | Finetuning |      | No                       |      | -      |      | 1.9      |      | 4.3      |      | 4.2      |      | 3.6      |
> | GLNN [7]   |      | No                       |      | -      |      | 1.9      |      | 3.9      |      | -        |      | 3.2      |
> | NOSMOG [8] |      | No                       |      | -      |      | 1.9      |      | 4.3      |      | -        |      | 3.6      |
> | EWC [9]    |      | Yes                      |      | -      |      | 2.3      |      | 4.0      |      | 4.2      |      | 6.1      |
> | MAS [10]   |      | Yes                      |      | -      |      | 1.8      |      | 3.9      |      | 8.7      |      | 8.2      |
> | TWP [11]   |      | Yes                      |      | -      |      | 16.3     |      | 3.1      |      | 7.4      |      | 5.2      |
> | LwF [12]   |      | Yes                      |      | -      |      | 1.9      |      | 4.3      |      | 4.2      |      | 3.6      |
> | **Ours**   |      | Yes                      |      | -      |      | **18.7** |      | **32.7** |      | **45.8** |      | **56.4** |

---

> ### Author Response · Authors · 2024-04-02
> **Response to Reviewer QCHX (Reference)**
>
> ### Reference
>
> [1] Hamilton and Ying et al., Inductive Representation Learning on Large Graphs, NeurIPS 2017
>
> [2] Huihui Liu, Yiding Yang, and Xinchao Wang. Overcoming catastrophic forgetting in graph neural networks. AAAI, 2021
>
> [3] Fan Zhou and Chengtai Cao. Overcoming catastrophic forgetting in graph neural networks with experience replay. AAAI, 2021
>
> [4] Chen Wang, Yuheng Qiu, Dasong Gao, and Sebastian Scherer. Lifelong graph learning. CVPR 2022
>
> [5] Zhang et al., CGLB: Benchmark Tasks for Continual Graph Learning, NeurIPS 2022
>
> [6] Liu et al., CaT: Balanced Continual Graph Learning with Graph Condensation, ICDM 2023
>
> [7] Graph-less neural networks: Teaching old mlps new tricks via distillation, Zhang et al. ICLR 2022.
>
> [8] Learning MLPs on Graphs: A Unified View of Effectiveness, Robustness, and Efficiency, Tian et al. ICLR 2023.
>
> [9] J. Kirkpatrick et al., “Overcoming catastrophic forgetting in neural networks,” CoRR, vol. abs/1612.00796, 2016.
>
> [10] R. Aljundi et al., “Memory aware synapses: Learning what (not) to forget,” in ECCV, 2018.
>
> [11] H. Liu et al., “Overcoming catastrophic forgetting in graph neural networks,” in AAAI, 2021
>
> [12] Z. Li and D. Hoiem, “Learning without forgetting,” TPAMI, 2018

---

> > ### Comment · Reviewer_QCHX · 2024-04-02
> > **Thank you**
> >
> > Thank you for the detailed reply and the additional explanations and experiments.

---

> > > ### Author Response · Authors · 2024-04-02
> > > **Thank you for your timely response**
> > >
> > > Thanks for your timely response! We would like to address them if you have any further concerns.

---

### Author Response · Authors · 2024-04-05
**Summary**

Dear Editor and Reviewers,

We sincerely appreciate your efforts in reviewing our paper and providing valuable feedback to enhance its quality. We have revised our manuscript in accordance with your suggestions and would like to summarize the modifications we have made.

We have discussed the motivation of our practical scenario in detail in Appendix A.4. Additionally, we have supplemented the introduction with further explanatory details.

In response to the suggestions, we have included additional baselines in our study. Specifically, we have incorporated two distillation-based graph learning models (GLNN, NOSMOG), three memory-free continual learning models (EWC, MAS, TWP), and three memory-based continual learning models (GEM, SSM, CaT), leading to a total of eight new baselines.

Furthermore, we have enriched our evaluation by introducing additional large-scale benchmarks, including CoraFull, Arxiv, Reddit, and Products. Our model, along with the newly introduced baselines, has been evaluated on these datasets. The new results further demonstrate the effectiveness of our model.

We hope the revisions could improve the overall quality and comprehensiveness of our manuscript. Thanks again for your helpful feedback!

Sincerely,

Authors of TMLR Paper 1966

---

### Decision · Action_Editor_7tKE · 2024-05-06

**Recommendation:** Accept as is

**Comment:**

The paper addresses a significant issue in the domain of class-incremental graph learning (CGL), where historical data may not be available during both training and inference due to constraints like storage or privacy regulations. This absence of historical data poses unique challenges, unlike in continual image classification where past data's unavailability affects only the training phase but not inference. The authors introduce ReplayGNN (ReGNN), a model designed to tackle catastrophic forgetting and the scarcity of neighbor information during inference without relying on memory buffers. The reviewers were leaning positive on the submission and appreciated the identification of previously unexplored setting. However, there were some concerns about experiment design and comparison with baselines. Post-rebuttal many of these were addressed by the additional experimental results, although some reviewers felt more comparisons might still be beneficial.

**Audience:**

Yes, the graph learning community will be interested in the paper

**Claims And Evidence:**

Yes